# Expression Profiling of Candidate Genes in Sugar Beet Leaves Treated with Leonardite-Based Biostimulant

**DOI:** 10.3390/ht8040018

**Published:** 2019-10-11

**Authors:** Hanifeh Seyed Hajizadeh, Bahram Heidari, Giovanni Bertoldo, Maria Cristina Della Lucia, Francesco Magro, Chiara Broccanello, Andrea Baglieri, Ivana Puglisi, Andrea Squartini, Giovanni Campagna, Giuseppe Concheri, Serenella Nardi, Piergiorgio Stevanato

**Affiliations:** 1Department of Horticulture, Faculty of Agriculture, University of Maragheh, Maragheh 55136-553, Iran; hajizade@maragheh.ac.ir; 2Department of Crop Production and Plant Breeding, School of Agriculture, Shiraz University, Shiraz 7144165186, Iran; bheidari@shirazu.ac.ir; 3Department of Agronomy, Food, Natural Resources, Animals and Environment, University of Padova, 35020 Legnaro (PD), Italy; giovanni.bertoldo.learning@gmail.com (G.B.); mcri.dellalucia@gmail.com (M.C.D.L.); chiarabr87@yahoo.it (C.B.); squart@unipd.it (A.S.); giuseppe.concheri@unipd.it (G.C.); serenella.nardi@unipd.it (S.N.); 4SIPCAM S.p.A., 20016 Pero (MI), Italy; fmagro@sipcam.com; 5Department of Agriculture, Food and Environment, University of Catania, 95131 Catania, Italy; abaglie@unict.it (A.B.); ipuglisi@unict.it (I.P.); 6COPROB, 40061 Minerbio (BO), Italy; giovanni.campagna@coprob.com

**Keywords:** biostimulant, BLACKJAK, gene expression profiling, humic acid, sugar beet

## Abstract

Leonardite-based biostimulants are a large class of compounds, including humic acid substances. Foliar application of biostimulants at field level improves plant growth, yield and quality through metabolic changes and stimulation of plant proton pumps. The present study aimed at identifying optimum dosage of BLACKJAK, a humic acid-based substance, which is able to modify genes involved in sugar beet growth. Thirty-three genes belonging to various biochemical pathway categories were tested in leaves of treated sugar beet (*Beta vulgaris* L.) samples to assess gene expression profiling in response to BLACKJAK. Seedlings of a diploid and multigerm variety were grown in plastic pots and sprayed with two dilutions of BLACKJAK (dilution 1:500–1.0 mg C L^−1^ and dilution 1:1000–0.5 mg C L^−1^). Leaf samples were collected after 24, 48, and 72 h treatment with BLACKJAK for each dilution. RNA was extracted and the quantification of gene expression was performed while using an OpenArray platform. Results of analysis of variance demonstrated that, 15 genes out of a total of 33 genes tested with OpenArray qPCR were significantly affected by treatment and exposure time. Analysis for annotation of gene products and pathways revealed that genes belonging to the mitochondrial respiratory pathways, nitrogen and hormone metabolisms, and nutrient uptake were up-regulated in the BLACKJAK treated samples. Among the up-regulated genes, Bv_PHT2;1 and Bv_GLN1 expression exerted a 2-fold change in 1:1000 and 1:500 BLACKJAK concentrations. Overall, the gene expression data in the BLACKJAK treated leaves demonstrated the induction of plant growth–related genes that were contributed almost to amino acid and nitrogen metabolism, plant defense system, and plant growth.

## 1. Introduction

Biostimulants affect plant responses to biotic and abiotic stresses by enhancing the nutrient uptake and by influencing the growth of roots and plants [1]. Different biostimulants proved to elicit morphological and physiological changes, improving the plant growth [2,3]. The positive effects of biostimulants on the plant growth encourage farmers to use them to reduce the reliance on fertilizers and pesticides. The European biostimulants industry council (EBIC) documented that over six million hectares of Europe agricultural farms have been treated with biostimulants since 2012 [4]. As stated by du Jardin, plant biostimulant is any substance or microrganism applied to plants with the aim to enhance nutrition efficiency, abiotic stress tolerance, and/or crop quality traits, regardless of its nutrients content [5].

Humic acid substances (HS) are abundant in the soil being the end products of microbial decompositions [1]. Proteins, carbohydrates, aliphatic biopolymers, and lignin represent the main compound classes in HS as well as the principal components in plants [6]. HS derivatives proved to enhance lateral roots and improve seedling root growth in over 16 plant species, including tomato [7], wheat [8], and maize [9]. Bean plants that were treated with HS were resistant to salt stress (120 mM NaCl) with respect to the control plants [10]. The effects of HS on salt and drought tolerances have also been demonstrated in other plant species [11,12]. Foliar spray of HS preparations increased photosynthetic pigment contents in eggplant [13] and the final yield in peanut [14], enhanced tolerance against abiotic stresses [15], improved phytohormone contents in snap bean [16], as well as the fresh and dry weight per plant in common bean [17].

Leonardite, which is an oxidized form of lignite enriched in HS, modifies physiological reactions in plants through transcriptional mechanism of the regulation of hormones and modifications in nitrogen and carbon metabolisms [8]. Although the application of leonardite-based biostimulants in the early stages of sugar beet growth demonstrated that these substances accelerate root growth and total sugar yield [18], their effects on the alteration of gene expression in sugar beet were investigated to a lesser extent. In tomato, the effects of a new biostimulant, EXPANDO^®^, containing different bioactive compounds, were tested at gene expression level, showing that the expression of about 4.000 genes, involved in several biological processes, such as transcription, stress responses, signal transduction, carbohydrate, and protein metabolism [19].

Specific functions and mechanisms of actions have not yet been determined for many biostimulant products [20]. The acquisition of a detailed knowledge of the molecular mechanisms underlying biostimulants helps to understand the biological functions of such substances. The aim of this study was to perform a high-throughput expression profiling of candidate genes in sugar beet leaves that were treated with leonardite-based biostimulant. The outcome of this study might assist sugar beet research community for a better understanding of the influence of biostimulants on metabolic changes for wide application of BLACKJAK at the field level.

## 2. Materials and Methods

### 2.1. Plant Materials and Growing Conditions

The sugar beet hybrid that was used in this study was the “Variety_1”, provided by the Department of Agronomy, Food, Natural Resources, Animals, and Environment—DAFNAE (University of Padova, Legnaro (PD), Italy). Variety_1 is a diploid and multigerm genotype showing homozygosity for resistance to rhizomania (*Rz1* gene) disease. This hybrid was a fingerprinted genotype selected from a preliminary genotypic screening in a previous study [21].

Seeds of 4 mm diameter size were soaked in 76% ethanol for 5 min, rinsed with sterilized water, and allocated on distilled water-moistened filter paper to germinate. Germination was carried out in a growth chamber at 25 °C in the dark for five days. Plants were grown on peat-based potting mix inside containers with depth-filtration system allowing gas exchange with limited dehydration and avoiding the dispersal of BLACKJAK treatments (Figure 1). Seedlings were cultivated in plastic pots kept in a climatic chamber at 25/18 °C and 70/90% relative humidity with a 14/10 h light/dark cycle (PPFD above shoot: 300 µE m^−2^ s^−1^). Plants were grown for 30 days and they were then sprayed with two dilutions of BLACKJAK (dilution 1:1000–0.5 mg C L^−1^ and dilution 1:500–1.0 mg C L^−1^). Leaf samples were collected 24, 48, and 72 h after the BLACKJAK treatment for each dilution. Three replicates were used for each treatment combinations. The new leonardite-formulate BLACKJAK used in the present study was a liquid commercial product provided by Sofbey (Puplinge, CH). Chemical, physical, and biological characteristics of leonardite-formulate BLACKJAK were described by Barone et al. [22]. The adopted dilution at farm level is 1:1000–0.5 mg C L^−1^. Sugar yield (t ha^−1^), sugar content, sodium, and potassium were determined for BLACKJAK untreated plants and those that were treated with 1:1000–0.5 mg C L^−1^ dilution.

### 2.2. Leaf RNA Isolation and cDNA Synthesis

Total RNA was extracted from 0.2 g of leaf tissues using a EurogoldTriFast kit (EuroClone, 360 Milan, Italy) and DNase-digested with TURBO DNA-free kit (Thermo Fisher Scientific, Waltham, MA, USA), according to manufacturer recommendations. The RNA was quantified by absorbance at 260 nm and Agilent 2100 Bioanalyzer checked RNA quality (Agilent Technologies, Palo Alto, CA, USA). Reverse transcription of RNA (500 ng) was achieved while using the SuperScript III reverse transcriptase kit (Thermo Fisher Scientific, Waltham, MA, USA), following the manufacturer’s instructions.

### 2.3. Real Time PCR analysis

The transcript levels of 33 sugar beet genes were analyzed (Table 1). Primers and probes were designed based on sequences that were selected from the reference genome RefBeet_1.2 by Thermo Fisher Scientific. Reactions for real-time PCR experiments consisted of a final volume of 5 µL containing 2.5 µL of 2× TaqMan Open Array master mix (Thermo Fisher Scientific, Waltham, MA, USA), and 2.5 µL of cDNA. Real-time PCR was analyzed on the QuantStudio 12K Flex Real-Time PCR System (Thermo Fisher Scientific) with respect to a thermocycler program as follow: 10 min pre-incubation at 95 °C, followed by 50 cycles of 15 s at 95 °C and 1 min at 60 °C. A negative and a positive control were routinely included, for each set of experiments. The melting curves post-PCR analyses were performed in order to estimate the specificity of the amplified products. Analysis of the genes relative expression was performed while using the comparative threshold Ct method [23,24]. The data for gene expression were normalized with respect to the arithmetic Ct mean of three housekeeping genes (*Tubulin*, Bv2_037220_rayf; *GAPDH*, Bv5_107870_ygnn; *Histone* H3, Bv6_127000_pera). The average Ct values from each sample of three housekeeping genes were converted into relative quantities while using the formula 2^−ΔCt^, where ΔCt was obtained after subtracting the lowest Ct value from the Ct value of the reference gene from the corresponding sample (ΔCt = each sample Ct value − the lowest Ct value) [25]. The relative expression (RE) for the expressed genes in response to BLACKJAK treatment was calculated while using the formula 2^−ΔΔCt^ where ΔΔCt was estimated by the difference between (Ct in target gene—average Ct in reference genes) for treatment and (Ct of the target gene—average Ct of the reference genes) in control.

### 2.4. Statistical Analysis

The mean for transcript abundance of the three housekeeping genes (Tubulin, GAPDH, and Histone H3) was used to normalize the data obtained. The Kyoto Encyclopedia of Genes and Genomes (KEGG) database [26] was used to search for the enzyme commission (EC) number of all assigned function related to a specific enzyme. Analysis of variance for factorial experimental design was performed in SAS software (SAS V 9.2 Institute, Cary, NC, USA). The data for gene expression heatmap were analyzed while using heatmapper software [27].

## 3. Results

Previous field trials at two locations in Italy were carried out to assess the consequences of BLACKJAK treatments on sugar content, yield, and quality (G. Campagna, pers. comm.). The results for agronomic traits demonstrated that BLACKJAK treated plants had higher sugar content and yield when compared with untreated plants whilst presented lower sodium and potassium (Table 2). Based on this evidence, the aim of this study was to perform an expression profiling of candidate genes in sugar beet leaves that were treated with BLACKJAK biostimulant for better understanding of its influence on sugar beet metabolic changes. The effects of different treatment dilutions were investigated for three exposure times (24 h, 48 h, and 72 h). Analysis of variance (ANOVA) showed that, among the 33 genes assayed with OpenArray qPCR, 15 had significant differences due to treatment, exposure time, and their interactions (Table 3). For eight genes, the main and interaction effects were significant. For the remainders, no significant effects for BLACKJAK treatment and exposure time were identified.

Figure 2 shows representative amplification curves from 12 selected genes. A large variation in their expression profiles is revealed from the different levels of the amplification curves. The plot for ΔRn versus PCR cycle associated with the expression of genes demonstrated the efficiency of the qPCR method used for differential response to treatment with BLACKJACK biostimulant (Figure 3). For each gene analyzed, a mean RE value was calculated by averaging all treatments and sampling times. Among the considered genes, the highest RE (1.4838) belonged to Bv_GLN1, whereas the lowest (0.00619) was identified for the EXO gene (Table 3). The data for RE demonstrated that the relative expression of most of the genes tested rose up as the exposure time with BLACKJACK increased. The relative expression of the genes assayed demonstrated that the number of up-regulated genes was increased as the concentration of BLACKJAK increased in the treated samples (Appendix A).

The expression of the Bv_UQCRH and Bv_AOAT1 genes increased as BLACKJAK exposure time increased in plant. Relative expression of these two genes in plants treated with 1:500 BLACKJAK after 72 h was higher than RE in other treatments (Figure 4). The gene Bv_AHG3 with the highest RE at BLACKJAK 1:1000 concentration after 72 h represented no direct relation with BLACKJAK concentration at the three time levels, whilst Bv_ATTIR1 was more expressed at a higher BLACKJAK concentration (Figure 5). The Bv_GLT1 and Bv_GLN1 genes were both more expressed as exposure time with BLACKJAK was elevated (Figure 6). Bv_PHT2;1 and Bv_PI4K showed higher expression in 1:1000 BLACKJAK concentration than in 1:500 BLACKJAK (Figure 7). In 1:1000 BLACKJAK treated samples, several genes were up-regulated at the end of experiment, of which Bv_PHT2;1 showed the highest RE.

The heat map for expression profiling in samples that were treated with 1:500 BLACKJAK showed that, except Bv_HAB1, the expression level of the tested genes was elevated as the exposure time was increased (Figure 8). The majority of genes were up-regulated 72 h after treatment with 1:500 BLACKJAK.

The pattern of expression profiling revealed that genes with differential expression are distributed between three exposure times in samples that were treated with 1:1000 BLACKJAK (Figure 9). The heat map revealed that the number of up-regulated genes increased as the BLACKJAK concentration increased. The highest number of up-regulated genes was identified in 72 h exposure time. Genes that were up-regulated in 24 h exposure time showed less expression in the remainder exposure time levels. The expression pattern of genes tested demonstrated that few genes showed up-regulation in two BLACKJAK treatments and exposure times.

## 4. Discussion

A better understanding of the effect of biostimulants in *Beta vulgaris* L. growth might be achieved through the knowledge of molecular mechanisms underpinning changes in gene expression in plant tissues. In the present study, the effect of various concentrations of BLACKJAK biostimulant on sugar yield, sugar content, and concentration of potassium and sodium ions was assessed in a sugar beet variety at different sampling times. Furthermore, several genes that belong to various categories (i.e., mitochondrial respiratory and signal transduction pathway, lipid, and hormone metabolisms) were selected for gene expression analysis in response to the treatment with BLACKJAK [28,29,30,31,32]. Treating sugar beet leaves with 1:1000 (0.5 mg C L^−1^) BLACKJAK resulted in higher sugar content (1.08%) and yield (6.78%) as compared with untreated plants. It has been demonstrated that biostimulants enhance processes such as photosynthesis, senescence, modulation of phytohormones, uptake of nutrients, resulting in higher performance and yield in plants [20]. In a study with sugar beet, the application of BLACKJAK increased sugar content (17.38%) and root yield than control demonstrating the positive effect of this biostimulant on sugar and plant growth [33]. The results of the current study demonstrated that the use of BLACKJAK as foliar spray reduced potassium (3.38%) and sodium (8.04%) components. Potassium and sodium ions are main components that make the sugar extraction difficult. Sodium and potassium both influence the pH of the raw sugar extract where alkalinity must be minimized for the efficient extraction of sugar.

The results of analysis of variance demonstrated that some of the evaluated genes significantly responded to treatment with BLACKJAK. Bv_UQCRH was up-regulated in leaves that were treated with 1:500 BLACKJAK, in only 72 h samples. Mechanism of action of humic acid-based biostimulants demonstrated that they enhance respiration and invertase activities making C substrates available [5]. The Bv_UQCRH gene encodes for an enzyme (cytochrome c reductase) belonging to the mitochondrial respiratory pathway and it is involved in mitochondrial electron transport, ATP synthesis, and cytochrome c reductase. Plant mitochondria act as “signaling organelles”, able to influence processes, such as nuclear gene expression. Respiration provides energy and carbon skeletons that are required for maintenance of existing dry matter [31]. Cytochrome c reductase plays a central role in the electron transport chain, because it collects reducing equivalents from NADH- and FADH_2_-linked dehydrogenases and passes them on to the terminal cytochrome system to form H_2_O. The increase in Bv_UQCRH expression level under the highest concentration of BLACKJAK tested could be effective for modulating the respiration to produce ATP instead of reactive oxygen species (ROS).

The results demonstrated that Bv_AOAT1 after 72 h exposure to 1:500 BLACKJAK concentration and Bv_GLT1and Bv_GLN1 in both concentrations were up-regulated. These three genes belong to the nitrogen metabolism category. The functions of these genes are associated with amino acid metabolism and glutamine synthetase (GS, EC 6.3.1.2). Glutamine synthetase is a central enzyme in nitrogen metabolism in leaves and a key component of nitrogen use efficiency and grain yield [33]. The modulation of GS expression and activity during a diurnal cycle, linked to carbon and nitrogen assimilation [34], is important for plant growth and development. In a study, Bao et al. [35] demonstrated that the GS-suppressed plants exhibited poor growth and were nitrogen transport deficient.

Three genes (Bv_AHG3, Bv_ATTIR1, Bv_PI4K) showing up-regulation in response to BLACKJAK treatment were associated with hormone metabolism and signal transduction pathways. Bv_AHG3 and Bv_ATTIR1, up-regulated after 48 and 72 h time exposure with BLACKJAK, are abscisic acid and auxin-related genes, respectively. Biostimulants that are developed from humic substances affect plant hormonal status [20]. It is possible that *de novo* synthesis of hormones might be induced in response to treating plants with humic based biostimulants. Amino acids, glycosides, polysaccharides, and organic acids are contained in many biostimulants and they may act as precursors or activators of endogenous plant hormones [20]. Abscisic acid (ABA) is involved in embryogenesis, seed physiology reactions, plant growth, and biotic and abiotic stress tolerance [36]. Young tissues have high ABA levels, although ABA has historically been considered a growth inhibitor [36,37]. It has been shown that ABA-deficient mutant plants are severely stunted and conversely exogenous ABA treatment of mutants restores normal cell expansion and growth [37]. The higher expression of Bv_AHG3 could be due to the activation of the abscisic acid hormone signal transduction pathway (Table 1). Annotation of Bv_ATTIR1, another up-regulated gene, was linked to auxin signal transduction. Auxins, a class of essential plant hormones, contribute to several plant growth and development processes, including cell elongation and cell division [38,39]. It has been shown that the growth regulator auxin contributed to both the initiation and elaboration of final morphology of leaf tissue and vascular networks [40]. Furthermore, in the model plants, *Arabidopsis*, rice and sorghum, the auxin-response genes (auxin/indole-3-acetic acid (Aux/IAA), auxin-response factor (ARF), and small auxin-up RNAs) are involved in growth/development and stress/defense responses [41]. Bv_PI4K was up-regulated in one of BLACKJAK levels assayed. This gene belongs to a signal transduction pathway category and it is involved in phosphoinositides (Pis) signaling. Studies of expression patterns of genes encoding key enzymes in PIs signalling have demonstrated that the expression of several enzymes is differentially regulated by various hormones and abiotic stressors, especially in the family of PIs phosphate kinase [42,43]. Besides their contribution to membrane building blocks, PIs exert regulatory effects on membrane polarity and plant growth [44]. Plants produce particular species of phosphoinositides under abiotic stress conditions through an increase in the degree of unsaturation in fatty acids [45,46]. The PIs associated pathways contribute to DNA replication, chromatin remodeling, response to environmental stresses, RNA export and cell cycle progression, hormone signaling, and plant defense network [47,48,49].

Bv_PHT2;1 expression exerted a three-fold change in 1:1000 BLACKJAK concentration whereas this gene was down-regulated in the other BLACKJAK treatment. This gene belongs to nutrient uptake category and it is involved in phosphate transport in plants. Plant growth is limited by the availability of P in most natural ecosystems. Phosphate transporters contribute to phosphate absorption and distribution, being important for physiologic reactions in plants [50]. In a study, *cis* elements were identified in the upstream regions of the phosphate transporter genes, many of which were related to growth or hormones [51].

The results of gene expression analysis showed that four genes (Bv_PEI, Bv_LAX1, Bv_LAX2, and Bv_MY10) were down-regulated as the exposure time with BLACKJAK increased in both 1:500 and 1:1000 treatments. The Bv_PEI belongs to lipid metabolism/degradation category. Several genes showed down-regulation as the exposure time was increased under one of BLACKJAK treatment only.

## 5. Conclusions

The results of the present study suggested that the genes belonging to various functional categories were differentially expressed in response to BLACKJAK and not all of the genes showed similar response to this leonardite-based substance. Few genes were up-regulated, demonstrating that BLACKJAK can stimulate specific pathways. Exposure time was an important factor for inducing higher gene expression in both BLACKJAK treatments. Increased expression level of mitochondria respiratory genes, glutamine synthase genes and the auxin and abscisic acid genes involved in signal transduction pathway might be an important mechanism explaining the enhanced growth. The possible contribution of Bv_GLN1 and Bv_PHT2;1, with higher expression in BLACKJAK treatments, to growth, stress tolerance, and disease resistance could be assessed in next studies. In conclusion, the results of our study provided information about genes and metabolic changes in response to BLACKJAK treatment in sugar beet that might assist sugar beet research community for better understanding the effects of biostimulants and, consequently, assure their wide use at the field level.

## Figures and Tables

**Figure 1 high-throughput-08-00018-f001:**
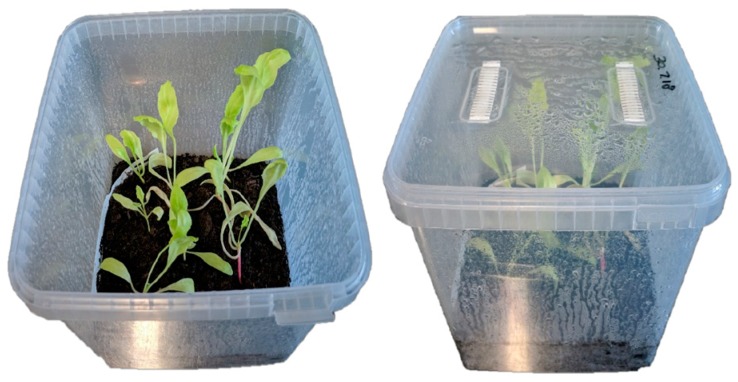
Plants grown on soil inside containers with depth-infiltration allowing for gas exchange with limited dehydration.

**Figure 2 high-throughput-08-00018-f002:**
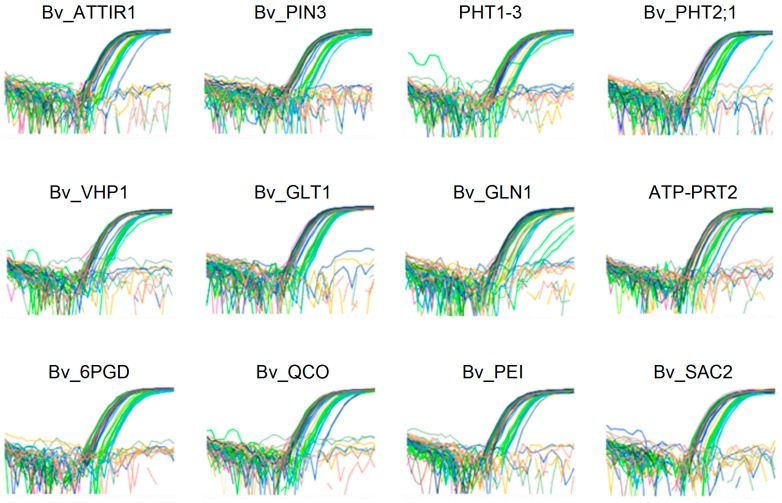
A representative amplification curve for the expression of several genes tested with OpenArray qPCR in sugar beet in response to treatment with BLACKJAK biostimulant.

**Figure 3 high-throughput-08-00018-f003:**
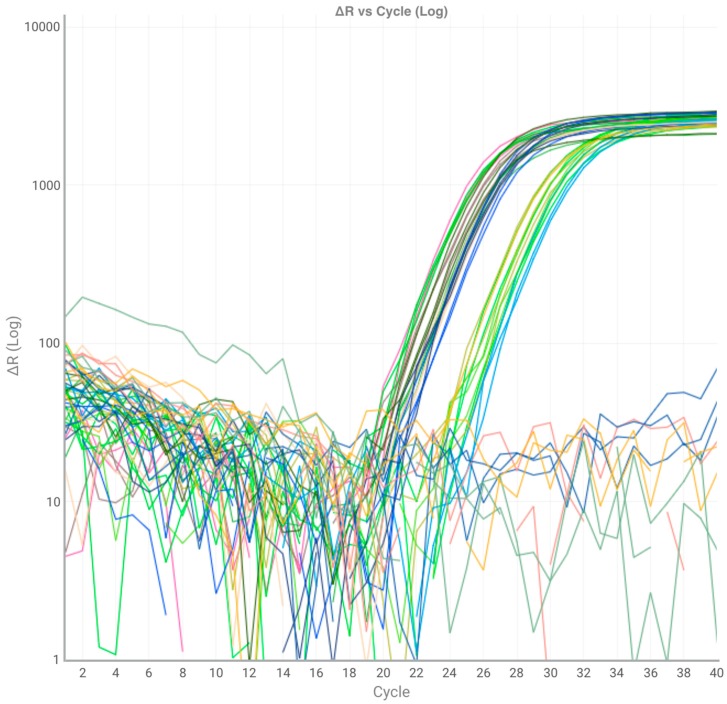
An amplification plot showing variations in ΔRn versus PCR cycles for the expression of genes tested in response to treatment with BLACKJACK in sugar beet leaves. Rn is the reporter signal normalized to the fluorescence signal dye.

**Figure 4 high-throughput-08-00018-f004:**
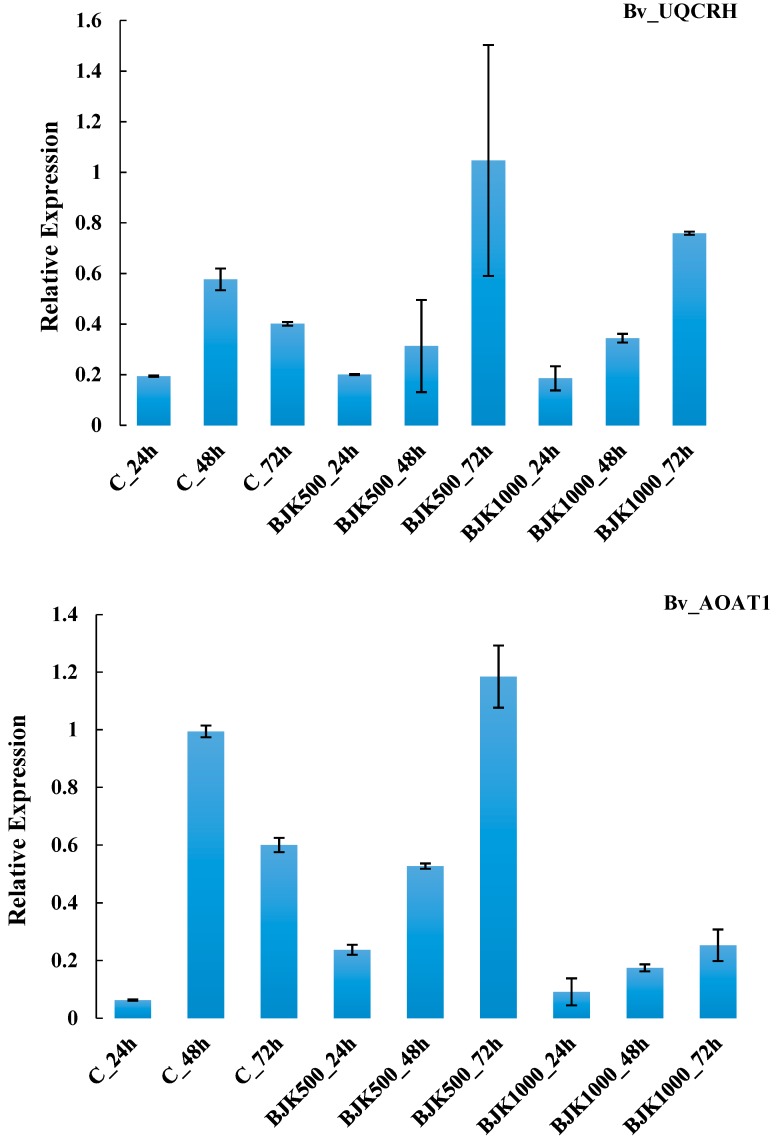
Relative expression of Bv_UQCRH and Bv_AOAT1 genes in BLACKJAK untreated and treated sugar beet in three exposure times. BJK: BLACKJAK, 500 and 1000 stand for 1.0 mg C L^−1^ and 0.5 mg C L^−1^ BLACKJAK, respectively.

**Figure 5 high-throughput-08-00018-f005:**
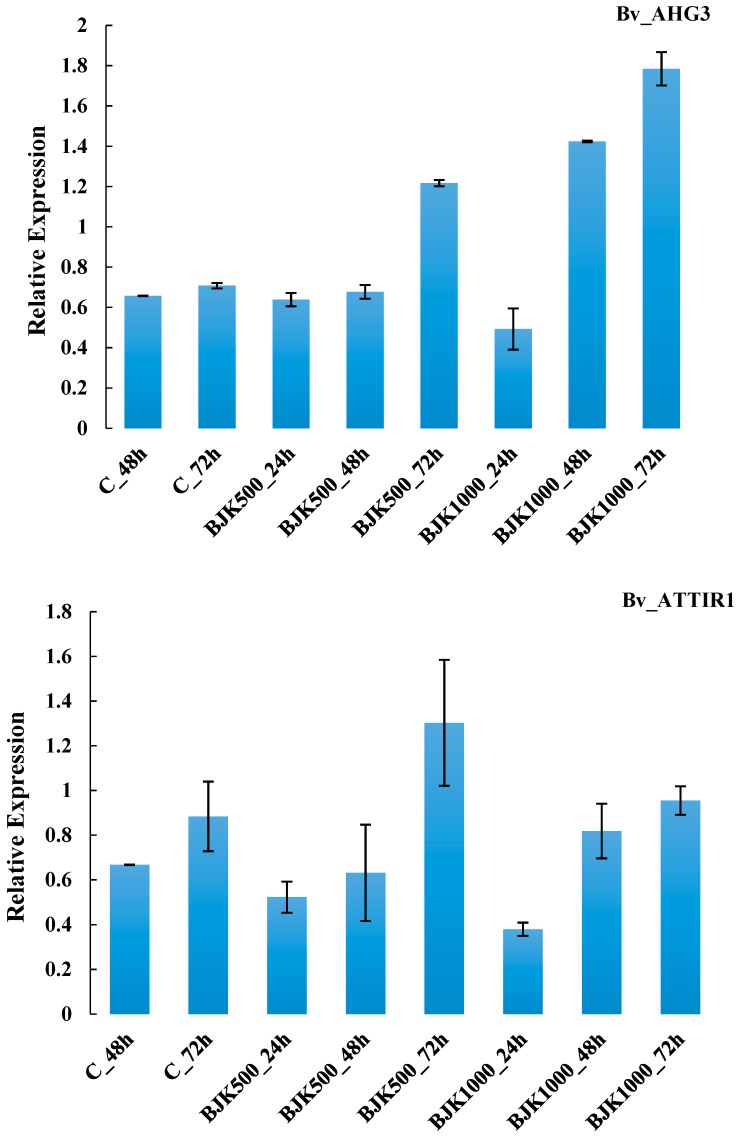
Relative expression of Bv_AHG3 and Bv_ATTIR1 genes in BLACKJAK untreated and treated sugar beet in three exposure times. BJK: BLACKJAK, 500 and 1000 stand for 1.0 mg C L^−1^ and 0.5 mg C L^−1^ BLACKJAK, respectively.

**Figure 6 high-throughput-08-00018-f006:**
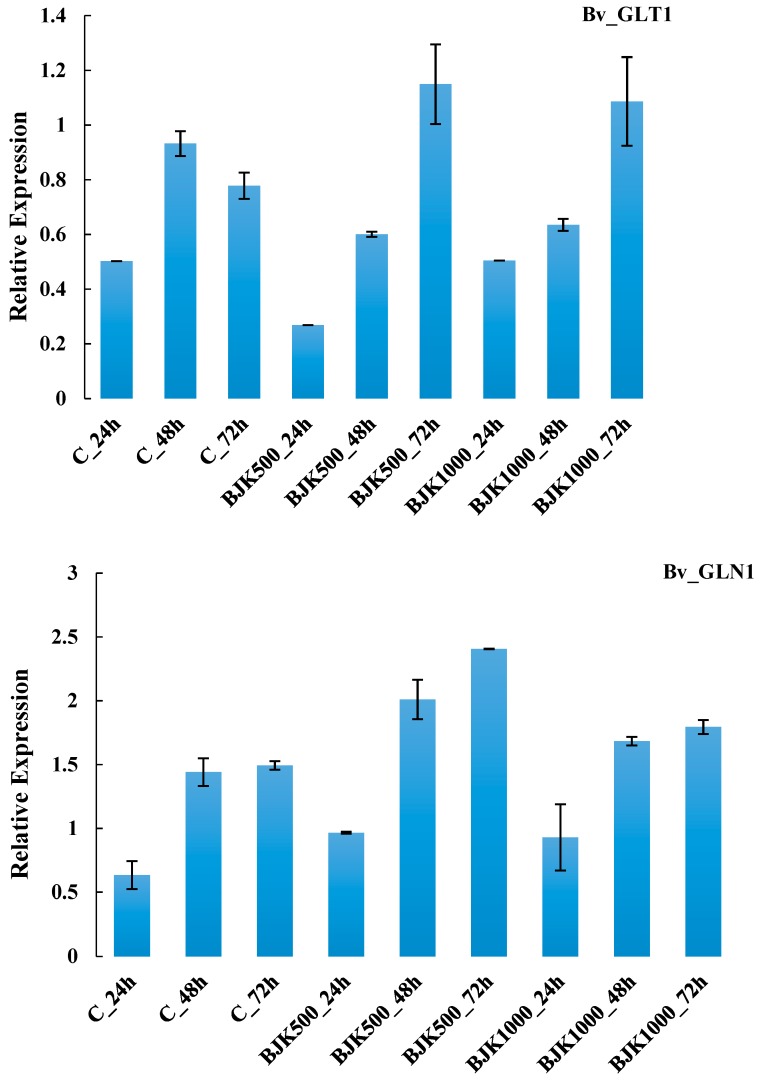
Relative expression of Bv_GLT1 and Bv_GLN1 genes in BLACKJAK untreated and treated sugar beet in three exposure times. BJK: BLACKJAK, 500 and 1000 stand for 1.0 mg C L^−1^ and 0.5 mg C L^−1^ BLACKJAK, respectively.

**Figure 7 high-throughput-08-00018-f007:**
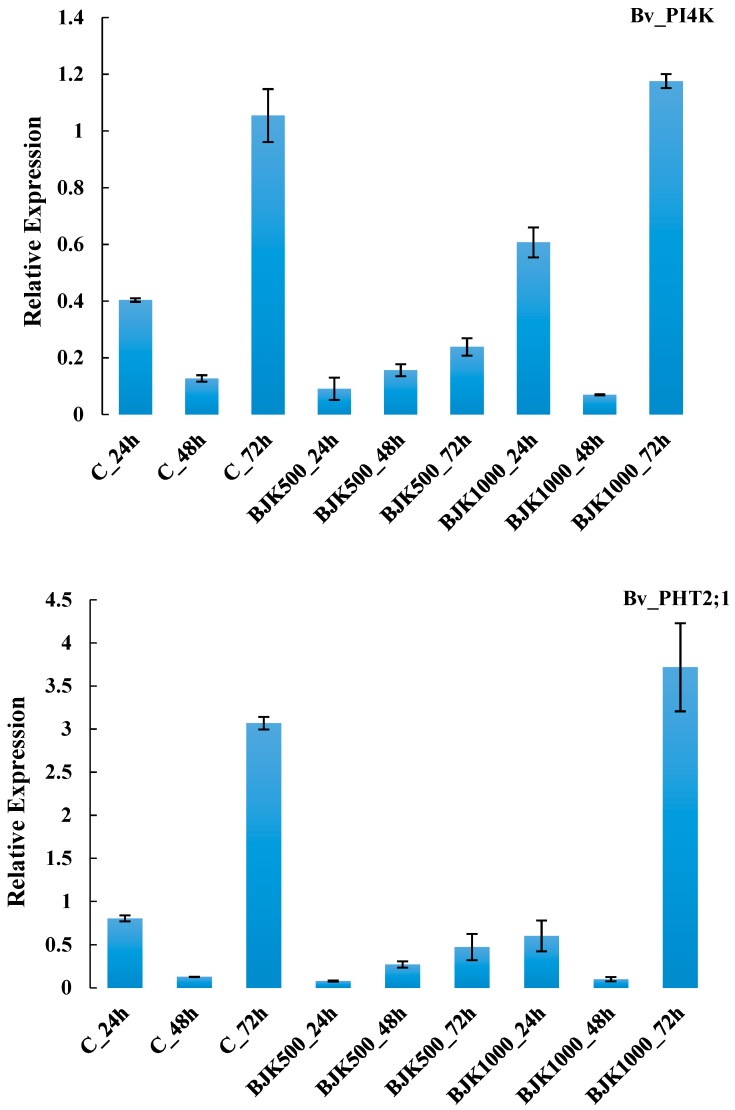
Relative expression of Bv_PIK4 and Bv_PHT2;1 genes in BLACKJAK untreated and treated sugar beet in three exposure times. BJK: BLACKJAK, 500 and 1000 stand for 1.0 mg C L^−1^ and 0.5 mg C L^−1^ BLACKJAK, respectively.

**Figure 8 high-throughput-08-00018-f008:**
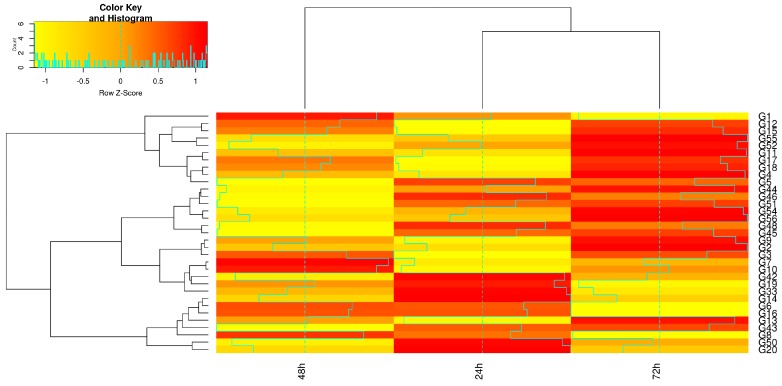
Heat map showing the expression profiling of genes in 1:500 BLACKJAK treated samples at three exposure times (h) in sugar beet leaf. Gene codes are referred to gene ID in Table 1.

**Figure 9 high-throughput-08-00018-f009:**
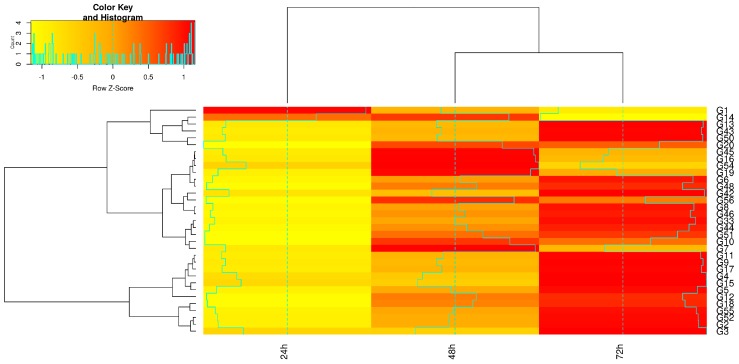
Heat map showing the expression profiling of genes in 1:1000 BLACKJAK treated samples at three exposure time (h) in sugar beet leaf. Gene codes are referred to gene ID in Table 1.

**Table 1 high-throughput-08-00018-t001:** List of genes tested for expression profiling in sugar beet plants.

Gene Code	Gene ID	Gene Function Category		EC Number
G2	Bv_6PGD	Mitocondrial respiratory pathway	OPP oxidative PP.6-phosphogluconate dehydrogenase	1.1.1.44
G3	Bv_R5PI		OPPnon-reductive PP.ribose 5-phosphate isomerase	5.3.1.6
G4	Bv_UQCRH		mitochondrial electron transport/ATP synthesis, cytochrome c reductase	1.10.2.2
G5	Bv_UQCR7		mitochondrial electron transport/ATP synthesis, cytochrome c reductase	1.10.2.2
G6	Bv_QCO		mitochondrial electron transport/ATP synthesis, cytochrome c	1.10.2.2
G7	Bv_PEI	Lipid metabolism	lipid metabolism, lipid degradation, lipases	3.1.1.3
G8	Bv_SAC2		lipid metabolism, lipid degradation, lysophospholipases	3.1.1.5
G9	Bv_AOAT1	N-metabolism	amino acid metabolism, central amino acid metabolism, alanine, alanine aminotransferase	2.6.1.2
G10	ATP-PRT2		amino acid metabolism, synthesis histidine, ATP phosphoribosyl transferase	2.4.2.17
G11	Bv_GLT1		N-metabolism, ammonia metabolism, glutamate synthase NADH dependent	1.4.1.14
G12	Bv_GLN1		N-metabolism, ammonia metabolism, glutamine synthetase	6.3.1.2
G13	Bv_AREB1	Hormone metabolism	hormone metabolism, abscisic acid, induced-regulated-responsive-activated	244.319.5
G14	Bv_HAB1		hormone metabolism, abscisic acid signal transduction	
G15	Bv_AHG3		hormone metabolism, abscisic acid signal transduction	
G16	Bv_LAX1		hormone metabolism, auxin signal transduction	
G17	Bv_ATTIR1		hormone metabolism, auxin signal transduction	
G18	Bv_PIN1		hormone metabolism, auxin signal transduction	
G19	Bv_LAX2		hormone metabolism, auxin signal transduction	
G20	Bv_PIN3		hormone metabolism, auxin signal transduction	
G21	Bv_SAUR		hormone metabolism, auxin induced-regulated-responsive-activated	
G22	Bv_CKX3		hormone metabolism, cytokinin synthesis-degradation	1.5.99.12
G23	Bv_AHK2		hormone metabolism, cytokinin signal transduction	
G24	Bv_CKDHase		hormone metabolism, cytokinin synthesis-degradation	1.5.99.12
G25	Bv_2OGD		hormone metabolism, gibberelin synthesis-degradation	
G26	Bv_GID1C		hormone metabolism, gibberelin signal transduction	
G27	Bv_GRP		hormone metabolism, gibberelin induced-regulated-responsive-activated	
G28	Bv_GAST1		hormone metabolism, gibberelin induced-regulated-responsive-activated	
G29	Bv_GASA14		hormone metabolism, gibberelin induced-regulated-responsive-activated	
G30	Bv_SAMT1		hormone metabolism, salicylic acid synthesis-degradation	
G31	Bv_SAMT2		hormone metabolism.salicylicacid.synthesis-degradation.synthesis.methyl-SA methyl esterase	
G32	Bv_HSP	Abiotic stress	heat stress	
G33	Bv_sah		heat stress	
G34	Bv_HPO3		cold stress	
G35	Bv_PMT21		drought/salt stresses	
G36	Bv_PMTI4		drought/salt stresses	
G37	Bv_TXN	Oxidative stress	redox thioredoxin	
G38	Bv_APX		redox ascorbate and glutathione ascorbate	______________
G39	Bv_GR		redox ascorbate and glutathione glutathione	______________
G40	Bv_FSD		redox dismutases and catalases	1.15.1.1/1.11.1.6
G41	Bv_CAT		redox dismutases and catalases	1.15.1.1/1.11.1.6
G42	Bv_CZSOD		redox dismutases and catalases	1.15.1.1/1.11.1.6
G43	Bv_CBR		redox misc	
G44	Bv_TIM	DNA synthesis/repair	protein targeting mitochondria	
G45	Bv_MY10		DNA synthesis/chromatin structure	
G46	Bv_DRT111		DNA repair	
G47	CRK27	Signal transduction pathway	signalling receptor kinases., leucine rich repeat I	
G48	IKU2		signalling receptorkinases, leucine rich repeat XI	
G51	SFH3		signallingin sugar and nutrient physiology	
G52	Bv_PI4K		signalling phosphinositides	
G49	ABCI6	Protein assembly	protein targeting secretory pathway	
G50	EXO		protein assembly and cofactor ligation	
G53	GABAt1	Nutrient uptake	transport amino acids	
G54	PHT1-3		transport phosphatase	
G55	Bv_PHT2;1		transport phosphatase	
G56	Bv_VHP1		transport H+ transporting pyrophosphatase	3.6.1.1

Note: EC: enzyme commission number.

**Table 2 high-throughput-08-00018-t002:** Mean of sugar content, yield, and processing quality related traits in BLACKJAK treated and untreated sugar beet.

BLACKJAK	Statistic	Trait			
		Sugar Content (%)	Sugar Yield (t ha^−1^)	Sodium (mmol 100g^−1^)	Potassium (mmol 100g^−1^)
Untreated	Mean	12.74	12.63	2.26	3.25
	STDEV	0.4	0.92	0.12	0.49
Treated	Mean	12.88	13.55	2.07	3.14
	STDEV	0.49	1.12	0.1	0.46

Note: STDEV: standard deviation.

**Table 3 high-throughput-08-00018-t003:** Results of analysis of variance (ANOVA) and relative expression analyses for genes tested with OpenArray qPCR in BLACKJAK treated sugar beet plants.

Code	Gene ID	Mean Squares				
		Treatment (T)	Exposure (E)	T × E	Error	Relative Expression (RE)
G2	Bv_6PGD	0.0043	0.0223 *	0.002	0.0035	0.1416
G3	Bv_R5PI	0.0052	0.0215	0.0045	0.007	0.1562
G4	Bv_UQCRH	0.0265	0.4461 **	0.112	0.0546	0.447
G5	Bv_UQCR7	0.081	0.0377	0.1093	0.0325	0.4909
G6	Bv_QCO	0.0008 **	0.0003 **	0.0002 **	0.00001	0.0316
G7	Bv_PEI	0.0005	0.0056	0.0006	0.0013	0.0873
G8	Bv_SAC2	0.0006 **	0.0003 **	0.0005 **	0.00001	0.0281
G9	Bv_AOAT1	0.3806 **	0.5031 **	0.2094 **	0.004	0.458
G10	ATP-PRT2	0.0036 **	0.0066 **	0.0032 **	0.00008	0.0995
G11	Bv_GLT1	0.0067	0.3501 **	0.8100 *	0.0174	0.7757
G12	Bv_GLN1	0.5487 **	1.9005 **	0.0561	0.0267	1.4838
G13	Bv_AREB1	0.00004	0.0002 *	0.00001	0.00003	0.0158
G14	Bv_HAB1	0.000001	0.00004 *	0.00001	0.000007	0.02499
G15	Bv_AHG3	0.0585	0.0427	0.3608 **	0.0339	0.9499
G16	Bv_LAX1	0.00005	0.00017	0.00006	0.00016	0.03032
G17	Bv_ATTIR1	0.0047	0.1016	0.0006	0.0397	0.7697
G18	Bv_PIN1	0.0023	0.0005	0.0258	0.0025	0.4049
G19	Bv_LAX2	0.000003	0.000005	0.000006	0.00007	0.0451
G20	Bv_PIN3	0.00000008	0.0000003	0.0000001	0.000001	0.0037
G33	Bv_sah	0.00000008	0.00000003	0.00000001	0.000001	0.0037
G42	Bv_CZSOD	0.0009 **	0.0057 **	0.0108 **	0.00003	0.0819
G43	Bv_CBR	0.0004 *	0.0016 **	0.0004 **	0.00006	0.0205
G44	Bv_TIM	0.0300 **	0.0655 **	0.0139 **	0.0007	0.139
G45	Bv_MY10	0.0031 **	0.0088 **	0.0042 **	0.00009	0.0691
G46	Bv_DRT111	0.0335 **	0.0241 **	0.0103 **	0.0012	0.1162
G48	IKU2	0.0091 **	0.0150 **	0.0058 **	0.0002	0.0854
G51	SFH3	0.0464 **	0.767 **	0.0233 **	0.0006	0.1764
G52	Bv_PI4K	0.3502 **	0.7668 **	0.1543 **	0.0033	0.436
G50	EXO	0.00001 **	0.00002 **	0.00004 **	0.000001	0.00619
G54	PHT1-3	0.0857 *	0.1624 **	0.0461	0.0166	0.1811
G55	Bv_PHT2;1	2.5789 **	8.8989 **	1.8073 **	0.0722	1.0261
G56	Bv_VHP1	0.0658 *	0.1954 **	0.0538 *	0.0098	0.1529

Note: * and ** stand for significant differences at 5 and 1% probability level.

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
