# Peer review of "Expression Profiling of Candidate Genes in Sugar Beet Leaves Treated with Leonardite-Based Biostimulant"

_2571-5135, 2019, doi:10.3390/ht8040018_

Round 1

Reviewer 1 Report

The manuscript proposed by Hajizadeh et al. It deals with the application of a product based on leonardite in the sugar beet crop and its ability to modify the relative expression of 33 genes. The results obtained are interesting, since they show overexpression of various genes related to different categories of crop metabolism, which can potentially increase growth and development, or the ability to tolerate some type of stress. However, some aspects of the manuscript must be improved.

Specific comments:

Abstract

Line 16. “Biosttimulants” must be “Biostimulants”

Introduction

Line 41. I suggest using the definition proposed by Du Jardin 2015 https://doi.org/10.1016/j.scienta.2015.09.021

Line 65. The doses used are not sufficient to meet the stated objective. I suggest modifying it according to the results presented.

Materials and methods

Line 85. How was the amount of water used in the control defined?

Line 86. Include the chemical, physical and biological characteristics of the product used.

Table 1. Review the Table as it seems incomplete, especially the categories: Hormone metabolism, Abiotic stress, and Oxidative stress.

Include at the bottom of the table the definition of asterisks.

Results

The main problem is that they do not present data on the growth and development of the crop, as well as data on the metabolites or compounds related to the genes analyzed. This would undoubtedly improve the quality of the manuscript.

Improve the quality of the Tables and Figures.

Line 127. A biplot is a representation of multivariate data, and what is shown in Figure 3 does not apply. What do they mean by a biplot?

Discussion

Expand the discussion. Only the genes that were overexpressed and the function they develop in the plant are counted. However, there is no explanation of the mode of action of the product evaluated.

Conclusions

Line 231. This should not be a conclusion, since the method is not the one being evaluated.

Line 242. With only two doses studied, an optimal dose cannot be concluded. I suggest modifying this conclusion.

Reviewer 2 Report

In the present study the authors present an interesting paper describing the expression profiling of 33 genes and their expression alteration under treatment with BLACKJAK – a leonardite-based biostimulant for 24, 48 and 72h. However, in the current state, the manuscript is largely descriptive and the results are under-interpreted. Thus, integration of up- and down-regulated transcripts among the highlighted metabolisms and with physiological data is necessary to infer their involvement with sugar beet growth.

Major remarks

Material and methods section

1- The authors indicate that BLACKJAK treatments improve plant growth, but their observation is based only on gene expression profiling. Changes in gene expression levels do not imply significant gain in the plant growth. The work lacks information about physiological parameters describing plant growth (fresh mass of leaves and of the entire plant, plant height, fresh mass and size of roots). These additional data should be presented and associated with gene expression profiling. This would give further evidences to the major conclusions of the manuscript.

2- Page 2, lines 79-84

The authors should specify in an ordered and clear way how seedlings and plants were cultivated. Were plants grown on a growth chamber after exposure to BLACKJAK? In which conditions of soil, light and temperature the plants were cultivated?

3- Page 3, lines 86-87

The authors should specify the standard concentration of the commercial leonardite-formulate (BLACKJAK) used in the dilutions.

4- Page 3, lines 99-114

It was not mentioned how primers and probes were designed? Besides, it is not described in this topic whether a final step including a melting curve and the usage of negative controls for qPCR essays were included. Authors should clarify this.

Results section

1- Results are under-interpreted and do not highlight the major findings. The authors should present the results in a more detailed manner integrating similar patterns of expression of genes from the same or different metabolisms and exposing the alterations from 24 to 72h of treatments.

2- qPCR results are better represented by relative expression values than fold change ratios (FC). Fold change ratio does not show the expression levels, it just expresses a ratio value between two comparisons. The replacement of tables and figures (heat maps) by bar chart of relative expression for each gene showing the values related to control, 24, 48, and 72h, for both dilutions in the same graph, would significant improve the visualization and interpretation of data. Here, the authors could include the statistical significance of relative quantities between control and the treatments.

Discussion section

Discussion is largely descriptive and needs to be significantly improved. The main findings should be better discussed and integration of major responsive genes highlighted. Besides, the discussion of the physiological data would significantly support the authors assumptions, previously based only on gene expression essays.

Minor remarks

Tittle

The tittle is misleading, the expression results of only 33 genes does not implies in “High-throughput expression profiling”.

Abstract

Page 1, line 16,the “biosttimulants” is misspelled.

Results section

Table 1 should include enzyme commission number (EC number) and a clear annotation of each Gene_Id.

Discussion section

1- Page 10, lines 180-181

The statement “Respiration provides energy and carbon skeletons are required for changes in respiration rate” is confused and should be revised.

2- Page 10, Lines 181-184

The final product of respiration is H2O not hydrogen peroxide (H2O2). This point should be revised.

Round 2

Reviewer 1 Report

The authors made the changes they had in the manuscript, which significantly improved the quality. The manuscript now has the quality to be accepted for publication in the current form.

Reviewer 2 Report

The authors have addressed most of my concerns in the previous manuscript. However I still have some points that need to be addressed.

Page 3, lines 106-122 - It still unclear whether a final step including a melting curve and the usage of negative controls for qPCR essays were included. Authors should clarify this.

Table 1 – The size font of the legend should be adjusted to that used in the main text. EC numbers are restricted to a few number of genes and a improved search using KEGG database would improved this point. Also, I missed the description of gene names, the given annotation/categorization is relevant, but too general and gene product is unclear. Thus EC number and the name of the gene should be specified.

The sentences highlighted by yellow commonly presents two words without separation. The authors should revise the main text.
